# Comprehensive Chemical Dust Suppressant Performance Evaluation and Optimization Method

**DOI:** 10.3390/ijerph19095617

**Published:** 2022-05-05

**Authors:** Ming Li, Rujia Wang, Gang Li, Xinzhu Song, Huaizhen Yang, Huinan Lai

**Affiliations:** 1School of Resources and Safety Engineering, Central South University, Changsha 410083, China; liming_csu@csu.edu.cn (M.L.); halosxz@163.com (X.S.); sdyanghuaizhen@163.com (H.Y.); lai_hui_nan@163.com (H.L.); 2Sinosteel Maanshan Mining Research Institute Co., Ltd., Maanshan 243000, China; hunankedaligang@163.com

**Keywords:** chemical dust suppression, dust suppression agent, dust pollution, evaluation method

## Abstract

Chemical dust suppression is an effective dust control technology. A dust suppressant component evaluation method that facilitates a complete selection of safe, efficient, and economical chemical materials has not been explored. Considering dust suppression performance, environmental safety, and cost-effectiveness of chemical dust suppressant technology, this study constructs a comprehensive evaluation index system of chemical dust suppressant performance, including the wetting performance, hygroscopic performance, bonding performance, annual cost per unit area, pH value of dust suppression solution, chemical toxicity, and chemical corrosion. Among them, the index characterizing the wetting performance of the solution is the sedimentation wetting time, which is determined by the dust sedimentation experiment; the index characterizing the hygroscopic performance of the solution is the evaporation stability time, which is determined by the evaporation experiment of the solution on the dust surface; the index to characterize the bonding performance of the solution is the surface wind erosion rate, which is determined by the wind erosion experiment of the solution on the dust surface; the toxicity of the solution is evaluated by the LD50 of the solution; the index to characterize the corrosion performance of the solution is the Q235 monthly steel corrosion rate, which is determined by the Q235 steel corrosion test. Corresponding evaluation parameters are determined including sedimentation wetting time, evaporation stabilization time, surface wind erosion rate; annual average use cost per unit area; solution pH value, chemical acute toxicity classification, monthly corrosion rate of Q235 steel, and corresponding standard test methods are also provided. In order to evaluate the comparability of the results, according to the specific requirements of the evaluation index system and the distribution characteristics of the measurement data, the data of each evaluation and detection index are standardized by linear transformation, range transformation and other methods, so that the obtained results are comparable. Considering the differences in the actual performance requirements of dust suppressants in different usage scenarios, the weights of evaluation indicators at all levels can be set independently and flexible. The experimental test data obtained through the example shows that: among the four chemicals selected to participate in the experiment, the comprehensive dust suppression performance score of Triton X-100 solution is in the poor-grade category. The comprehensive dust suppression performances of calcium chloride solution, water, and polyacrylamide solution scored high in the average-grade category. The comprehensive evaluation process is logically correct, and the results are consistent with the phenomena observed in the experiment, consistent with conventional understanding, and have strong credibility. This method can provide a standardized evaluation technique and test process for the comprehensive performance evaluation and comparison of chemical materials and dust suppressants.

## 1. Introduction

Dust pollution is a major environmental problem and the main cause of occupational health hazards [1]. In actual production, dust will have a certain impact on the environment and personnel health. The work of Dongyue Li and Yilan Liao proved that railway coal transportation has a certain effect on heavy metals in street dust [2], Huang et al. studied the effects of different concentrations of PM2.5 exposure on the cardiopulmonary function of manganese mine workers [3]. The results showed that PM2.5 exposure caused damage to the lung function of open-pit manganese mine workers, and restrictive ventilation disorders were the most common, Maasago M. Sepadi et al. studies have identified increased health risks for miners due to chronic low levels of dust exposure and lack of use of RPE (respiratory protective equipment) [4]. Not only the health of workers who have direct contact with industries that generate dust is affected, but also the health of people living near mines and quarries due to environmental pollution caused by dust. Samantha Iyaloo et al. studied the respiratory health of a community living near a gold mine waste dump and showed that residents living within 500 m of the mine had elevated adverse respiratory effects [5]. Respiratory and ocular symptoms and objective measures of respiratory disease were higher among the most exposed groups of study participants; Maysaa Nemer et al. studied the lung function and respiratory health of residents near quarries in Palestine [6], exploring the negative health effects of environmental dust exposure in two communities near quarries in Palestine.

As an effective dust control technology, chemical dust suppressants have been widely studied and applied [7]. Therefore, various new dust suppressants have been developed. Zhang et al. developed an environmentally friendly dust suppressant with improved wettability and solidification performance. The results of spray dust removal experiments show that the developed spray dust reducer can significantly reduce the dust concentration. The average dust removal rates of total dust and respirable dust increased to 83.94% and 84.08%, respectively [8]. Qiu et al. prepared a new dust suppression gel by graft copolymerization of itaconic acid-acrylic acid polymer and bentonite. The prepared dust-suppressing gel is suitable for dustproofing during production and transportation in the coal mine industry, and can effectively reduce the water consumption during dust-proofing under the condition of improving dust-removing efficiency [9]. In industrial production, there are pollution characteristics of dust generation, and the requirements for dust suppressants are diverse. In addition, the chemical materials of dust suppressants vary extensively. How to effectively select and evaluate the dust suppression performance of different chemical materials based on the various demands for dust prevention and control requires urgent attention.

Evaluation indexes researched by Wu et al. for the performance evaluation of prepared dust suppressants include sedimentation performance, solution pH, wind erosion resistance, corrosion, toxicity, and economic benefits [10]. Zhou et al. analyzed the infrared spectrum of lignite dust in a performance evaluation of a composite wetting dust remover and studied the wetting ability of surfactants, the atomization performance of droplets, and the influence of inorganic salt on water evaporation rate [11]. Chen et al. analyzed water retention, pressure resistance, rain erosion resistance, fluidity, and permeability in the performance test of a new liquid dust suppressant, but did not compare it with the comprehensive dust suppression performance of other dust suppressants [12]. Yanqiang LI proposed that the evaluation indexes of chemical dust suppressants should include corrosion, toxicity, environmental pollution, and cost performance [13]. Chen et al. constructed an optimization decision-making model of inhibitors with the evaluation indexes of pressure resistance, wind erosion resistance, frost resistance, temperature resistance, water retention, and viscosity [14]; however, they only considered the effect of dust suppressants on dust suppression performance. To the best of our knowledge, neither a multi-angle, comprehensive dust suppression performance evaluation method that is generally applicable to chemicals nor a widely recognized comprehensive evaluation index system of dust suppressants has been proposed. Thus, this study investigates a comprehensive performance evaluation method of chemical dust suppression technology from the aspects of dust suppression mechanisms, economic benefit, and environmental safety.

## 2. Basic Principles of the Optimization and Evaluation Index System

The main mechanism of chemical dust suppression can be divided into wetting, moisture absorption, bonding, and composite effect. As the basic requirement of dust suppressant performance evaluation, the aforementioned performance components should be included in the evaluation index system of the comprehensive performance of chemical material dust suppressants. As an artificial material additive, the application of the chemical dust suppressant should comply with the environmental requirements of the use site, including environmental toxicity, corrosion, and acid-based properties. On this basis, the production cost and economic benefits of different chemical materials should be compared. In summary, the comprehensive performance evaluation of chemical dust suppression technology should include dust suppression performance, environmental safety, and economic impact, as shown in Figure 1.

Based on the three first-level indicators of the comprehensive evaluation, corresponding second-level indicators are selected, as shown in Figure 1. Dust suppression performance comprises the wetting performance, hygroscopic performance, and bond performance. The economic indicator is the comprehensive use cost. Indicators of environmental safety are the pH value of the dust suppression solution, chemical toxicity, and chemical corrosion. Specific evaluation parameters were further determined: deposition wetting time, evaporation stabilization time, surface wind erosion rate, the annual average cost per unit area, pH value of dust suppression solution, chemical acute toxicity classification, and monthly corrosion rate of Q235 steel.

## 3. Evaluation Parameter Test Method

For obtaining comparable quantitative evaluation parameter values, the corresponding standardized test method and workflow were reasonably determined according to each evaluation parameter.

Sedimentation wetting time: (a) The test solution is configured according to the mass concentration of 3% and slowly poured into a 25-mL test tube until the scale line reaches 25 mL. In actual production and life, the choice of dust suppressant concentration is mostly about 1–5%. In this experiment, considering the limitation of the comprehensive dust suppression ability of a single chemical reagent, the median concentration of 3% is selected as the test concentration in the experiment. Through the choice of the concentration of this solution, the experimental phenomenon can be more obvious. At the same time, the experimental time can be appropriately shortened, which is more convenient for the analysis of the subsequent experimental measurement data. (b) A dried 1-g test dust sample is gently placed into the solution of the test tube. The time required for all of the dust particles to settle at the bottom of the solution is recorded. The experiment is repeated three times, and the average time is recorded as the sedimentation wetting time. Figure 2 is the schematic diagram of the sedimentation wetting time experiment.


Figure 2Schematic diagram of sedimentation wetting time experiment.
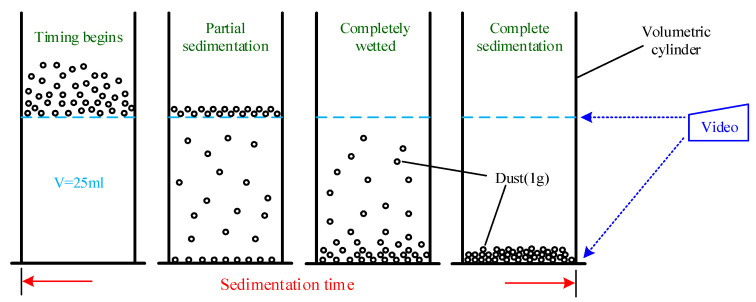



2.Evaporation stabilization time: (a) Dry 50-g test dust samples are stacked in a φ100-mm glass petri dish, showing a natural accumulation. (b) A total of 10 mL of the test solution is uniformly dropped with 3% mass concentration on the surface of the test dust reactor. The initial mass is recorded after 10 min. (c) The test dust sample is placed in a drying oven (50 °C, windless), and the dust mass is weighed and recorded every 10 min until the mass change rate is less than 0.1%. The evaporation stabilization time is recorded. Figure 3 is the schematic diagram of the stabilization experiment.

3.Surface wind erosion rate: (a) The dry 50-g quartz dust samples are naturally stacked in a φ100-mm glass surface dish, and the dust surface is gently scraped. (b) Then, 10 mL of the test solution with a mass concentration of 3% is evenly dropped on the surface of the test dust pile. The initial mass *m*_0_ is weighed and recorded after 10 min. (c) The test dust sample surface plate is placed in a stable flow field (wind speed 4 m/s) and fixed horizontally. The final mass *m*_1_ is weighed and recorded after 20 min of placement in the flow field. (d) The surface wind erosion rate is calculated according to the following formula: (*m*_1_ − *m*_0_)/*m*_0_ × 100%. Figure 4 is the schematic diagram of surface wind erosion rate experiment.

4.Annual average use cost per unit area: material cost, equipment cost, and operation cost. (a) Material cost *C*_1_: The effective action time of a single dust suppressant is *t* days, the single-use cost is *c* yuan, and the value is 360 days per year. (b) Complementary equipment cost *C*_2_: For *m* sets of required equipment, the service life of the first equipment is *n_i_* years, the purchase cost of equipment is *x_i_* yuan, the transportation and installation cost is *y_i_* yuan, the maintenance time is *w_i_*, and the average single maintenance cost (e.g., labor costs) is *z_i_* yuan per time. (c) Operation cost *C*_3_: The spraying period of dust suppressant is set to *T* days, and *n* devices are necessary for single spraying of dust suppressant. The *i*-th device is used for *ti* hours, and the power of the device is *p_i_* kW. The unit price is *a* yuan per degree. A single use manually requires *x* people, labor time is *y* yuan per day, and working time is *z* days. (d) Assuming that the single-use area is *S* m^2^, the annual average use cost *C* per unit area is calculated according to Equation (1).


(1)
$$\left\{ \begin{array}{l} C_{1}=\frac{360}{t}\times c\\ C_{2}=\sum_{i=1}^{m}\frac{x_{i}+y_{i}+w_{i}z_{i}}{n_{i}}\\ C_{3}=\frac{360}{T}\times (x\cdot y\cdot z+\sum_{i=1}^{n}t_{i}\cdot p_{i}\cdot a)\\ C=\frac{360\cdot c}{t\times S}+\sum_{i=1}^{m}\frac{x_{i}+y_{i}+w_{i}z_{i}}{n_{i}\times S}+\frac{360}{T\times S}\times (x\cdot y\cdot z+\sum_{i=1}^{n}t_{i}\cdot p_{i}\cdot a) \end{array}\right.$$


5.pH value of dust suppression solution: After the dust suppression solution is fully stirred and stable, the pH value is detected by pH test paper or a pH meter, repeated three times, and the average value is taken.6.Acute toxicity classification of chemicals: LD50 data were classified according to acute toxicity classification criteria of the World Health Organization (WHO) using acute oral LD50 values of mice (Table 1). For chemicals without data sources, standardized experimental tests were conducted according to the relevant requirements of “Technical specification for chemical toxicity identification” to obtain relevant toxicity grading data.

7.The monthly corrosion rate of Q235 steel: (a) Experiments were conducted in accordance with the test methods and test process control requirements of the “Cyclic immersion test of corrosive salt solution for metals and alloys” (GB/T 19746-2018). (b) Q235 steel with a rectangular, thin plate (90 mm × 120 mm × 2 mm) was used as the test block, the initial mass was weighed, and *m*_0_ was recorded after cleaning and drying. The test block was completely immersed in the test solution with a mass concentration of 3% and placed in a constant temperature- and humidity-controlled (20 °C, 90%) experimental box for 30 days. During the experiment, the test block remained suspended in the test solution. (c) The test block was carefully removed from the solution, the corrosion products of the test block were removed according to the requirements of “Elimination of corrosion products on corrosion specimens of metals and alloys” (GBT 16545-2015), and the mass of the test block was weighed after the corrosion products were removed by washing and drying. (d) The monthly corrosion rate of the Q235 steel was calculated according to the formula (*m*_1_ − *m*_0_)/*m*_0_ × 100%. Figure 5 is the figure of the monthly corrosion rate of Q235 steel experiment.

## 4. Evaluation Parameter Weight and Standardization Processing

### 4.1. Evaluation Workflow

The weight of the evaluation parameters of the comprehensive performance of chemical dust suppression should be determined according to the actual use and environmental context. For example, using dust suppressants in urban areas should appropriately increase the weight of environmental safety, the cohesive weight should be increased in the field with a large wind speed, and the wet weight ratio should be considered for hydrophobic dust. In the actual operation process, the expert scoring method can be used to determine the weight value of each evaluation parameter. The basic process of the comprehensive evaluation is displayed in Figure 6.

First, whether the chemical fulfills the basic safety requirements were judged; that is, the pH limit range of the dust suppression solution is set to be 4.0 ≤ pH ≤ 10.0, and the oral acute toxicity of the chemical is not higher than grade 3 (low toxicity), that is, LD50 ≥ 501 mg/kg. If either of the aforementioned two indicators exceeds the limit, the chemical is determined to be unsuitable for use as a dust suppressant component. The evaluation parameter values corresponding to the second-level evaluation indexes of chemical dust suppressants were obtained (see Figure 1). If all the evaluation parameter values did not exceed the set limit, the score values of each index in the second-level evaluation index were obtained according to the standardization of the evaluation parameter values, and the basic principle of optimization and the total score of each index in the first-level evaluation index in the evaluation index system was obtained by weighted calculation.

### 4.2. Standardization of Evaluation Parameters

Commonly used evaluation parameter standardization methods include the range transformation method and linear proportional transformation method [15]. In the proposed optimization method, considering that various evaluation parameters are included, the rules for the parameters differ. In this study, the standardized values are divided into three grades. The excellent grade is (80, 100], the average grade is [50, 80), and the poor grade is [0, 50). Thus, different standardization methods are adopted for different evaluation indexes to make them dimensionless, and the value range is [0–100].

The measurement values of each evaluation parameter of *i* chemicals are recorded as follows: sedimentation wetting time (*A_i_*), evaporation stability time (*B_i_*), surface wind erosion rate (*C_i_*), annual average use cost per unit area (*D_i_*), dust suppression solution pH value (*E_i_*), mice oral acute LD50 (*F_i_*), and Q235 steel monthly corrosion rate (*G_i_*).

Ranges of the evaluation parameters are set as follows: sedimentation wetting time (*A*_min_–*A*_max_), evaporation stabilization time (*B*_min_–*B*_max_), surface wind erosion rate (*C*_min_–*C*_max_), annual average use cost per unit area (*D*_min_–*D*_max_), pH value of dust suppression solution (*E*_min_–*E*_max_), acute oral LD50 toxicity grade of mice (*F*_min_–*F*_max*)*_, and the monthly corrosion rate of Q235 steel (*G*_min_–*G*_max_*)*.

To facilitate the function expression, the lower limit of each evaluation parameter is *S*_min_, the upper limit is *S*_max_, and the measurement value of the evaluation parameter is *S_i_*. According to the different change rules of each evaluation parameter, the classification standardization process is conducted, and the initial evaluation value *Z_i_* value range is [0–100].

Standardization of sedimentation wetting time evaluation parameters. This type of evaluation index is a reverse index. The smaller the value of the evaluation parameter, the better the performance, and the standard deviation of the measured value is large, which is not suitable for direct linear ratio transformation. Therefore, the standardization method combining the linear ratio and the range transformation method is adopted. The *y_(i)_* represents the measured value of the evaluation parameter of the *i*th chemical after transformation using a linear scale. The maximum and minimum values are *y*_max_ and *y*_min_, respectively. *Z_Ai_* is the standard value of the sedimentation wetting time index of the *i*th chemical
(2)$$y_{\left(i\right)}=\frac{S_{\min}}{S_{i}}$$
(3)$$Z_{A_{i}}=\frac{y_{i}-y_{\min}}{y_{\max}-y_{\min}}\times 100$$

Standardization of the measured values of evaporation stabilization time evaluation parameters. This type of evaluation index is a positive index—the greater the value of the evaluation parameters, the better the performance—and the standard deviation of the measurement value is small. If the range transformation method is used, the dispersion of the evaluation results is high. Therefore, the linear proportional transformation method is used to standardize the measurement value of this type of evaluation index and multiply the efficacy coefficient [16] to make the range of the standardized value normal. The formula $\overset{}{\overset{-}{\underset{}{S}}}$ is the average value of the measured value of the evaluation parameter. *Z**_B_**_i_* is the standard value of the evaporation stabilization time index of the *i*th chemical.



(4)
$$Z_{B_{i}}=\frac{1-e^{-\frac{S_{i}}{\bar{S}}}}{1+e^{-\frac{S_{i}}{\bar{S}}}}\cdot \frac{S_{i}}{S_{\max}}\times 100$$



Standardization of surface wind erosion rate and monthly corrosion rate of Q235 steel. This type of evaluation index is a reverse index. The smaller the evaluation parameter value, the better the performance, and the standard deviation of the measurement value is small. If the range transformation method is used, the dispersion of the evaluation results is high. Therefore, the linear proportional transformation method is used to standardize the measurement value of this type of evaluation index. *Z**_C_**_i_* is the standard value of the surface wind erosion rate index of the *i*th chemical. *Z**_G_**_i_* is the standard value of the monthly corrosion rate of Q235 steel index of the *i*th chemical.


(5)
$$Z_{C_{i},G_{i}}=\frac{S_{\min}}{S_{i}}\times 100$$


Standardization of solution acidity and alkalinity evaluation parameters. The pH value of the solution is the evaluation index, which ranges from 0 to 14, and the pH value is the best when it is 7. Therefore, the standardization of this index is the standardization of appropriate indicators; that is, the closer the measured value of the evaluation parameters is to the appropriate value, the better. *Z**_E_**_i_* is the standard value of the pH value index of the *i*th chemical.


(6)
$$Z_{E_{i}}=\frac{100}{\sqrt{2\pi }}\cdot \exp^{-{\left(S_{i}-7\right)}^{2}}$$


Standardization of annual average cost per unit area and evaluation parameters of acute oral LD50 toxicity grading in mice. This type of evaluation index is a reverse index—the smaller the evaluation parameters, the better the performance—and the standard deviation of the measured value is large, which is not suitable for direct linear scale transformation. Therefore, the range transformation method is used to standardize it. The LD50 data were classified according to the WHO acute toxicity grading standard, and the relevant toxicity grade was 1–6. *Z**_D_**_i_* is the standard value of the annual average cost per unit area index of the *i*th chemical. *Z**_F_**_i_* is the standard value of the acute oral LD50 toxicity grading in mice index of the *i*th chemical.


(7)
$$Z_{D_{i},F_{i}}=\frac{S_{\max}-S_{i}}{S_{\max}-S_{\min}}\times 100$$


### 4.3. Comprehensive Performance Evaluation Analysis

According to the evaluation index system in Figure 1, the weight of the first level index is denoted as *W*_I*i*_ (*i* = 1–3), the weight of the second-level index is denoted as *W*_II*j*_ (*j* = 1–7), and the weight of the evaluation parameter is denoted as *W*_III*j*_ (*j* = 1–7). The total weight of ownership of each level index is 1. *Z_ij_* is the standardized score of the *j*th evaluation parameter of the *i*th chemical, and *V_i_* is the total standardized score of comprehensive performance. Then, the standardized comprehensive score calculation formula of the *i*th chemical is as follows:(8)$$V_{i}=\sum_{j=1}^{7}W_{\mathrm{III}j}\times Z_{ij}$$

The standardized score range of comprehensive evaluation obtained by calculation is [0, 100]. The larger the value of the comprehensive evaluation score, the better the comprehensive dust suppression performance. The standardized score range of the comprehensive evaluation can also be divided into three grades. The excellent grade is (80, 100], the average grade is [50, 80), and the poor grade is [0, 50).

## 5. Case Analysis

According to the aforementioned evaluation methods, calcium chloride, Triton X-100, polyacrylamide, and sodium hydroxide (i.e., four types of chemical materials) were selected for example analysis, and the water solution was established as the control. According to the aforementioned test data, the first step was to assess whether the chemicals fulfilled the basic safety requirements: sodium hydroxide LD50 was 40 mg/kg, which is more than the limit, and thus was neither suitable for a dust suppression agent nor analyzed in the subsequent experiments. Figure 7 shows the images of the indoor experiment. The weight coefficient of the evaluation parameters is obtained by the expert scoring method, and the results are shown in Table 2.

Figure 7a shows the sedimentation time experiments and data of the three chemicals purified water, calcium chloride, and Triton X-100. Because the sedimentation time of polyacrylamide is too long, it is not convenient to display in the figure. Experiments show that the wetting properties of the four chemical solutions are ranked from strong to weak: triton X-100, purified water, and polyacrylamide.

Figure 7b shows the experimental phenomena and data of the surface weathering rate of purified water, calcium chloride, Triton X-100 and polyacrylamide and the monthly corrosion rate of Q235 steel. It can be clearly seen from the experimental phenomenon of surface wind erosion rate that the experimental group of polyacrylamide can see that the dust surface is covered with liquid after the experiment is completed, while the other groups cannot see the trace of liquid covering, which proves that the wind erosion rate of polyacrylamide is the smallest. The combined data chart shows that the adhesion properties of the four chemical solutions are polyacrylamide, calcium chloride, purified water, and Triton X-100 from strong to weak. It can be seen from the Q235 month steel corrosion rate experimental phenomenon diagram that the Triton X-100 experimental group has the most obvious corrosion marks on the surface of the steel sheet, indicating that its corrosion resistance is the weakest. Combined with the data chart, it is shown that the corrosion resistance of the four chemical solutions is polyacrylamide, calcium chloride, purified water, and Triton X-100 from strong to weak.

Figure 7c shows the experimental phenomenon diagram and data diagram of the evaporation stability experiment. The results show that Triton X-100 has the longest time to achieve relative stability in evaporation, and polyacrylamide achieves evaporation stability the fastest. The hygroscopic properties of the four chemical solutions from strong to weak are Triton X-100, calcium chloride, purified water, and polyacrylamide.

According to the specific data of the evaluation parameters obtained in the experiment, the standard values obtained after processing according to the standardization method of Section 4.2 are shown in Table 3, where the * symbols indicate the reference values of the literature [17].

According to the measurement experiments in Section 3, the selected chemicals are experimentally measured, and the specific measurement data obtained are shown in Table 3. Standardize the data in Table 3 according to the above evaluation process in Section 4.2, for example, the calculation examples of each evaluation parameter of the aqueous solution are as follows:

Sedimentation wetting time:(9)ZAwater=y(water)−y(min)y(max)−y(min)×100=SminSwater−SminSmax1−SminSmax×100=278.3632−278.31257.61−278.31257.6×100=28.2

Evaporation stabilization time:(10)$$\bar{S}=\frac{540+610+580+510}{4}=560$$
(11)$$Z_{B_{water}}=\frac{1-e^{-\frac{S_{water}}{\bar{S}}}}{1+e^{-\frac{S_{water}}{\bar{S}}}}\cdot \frac{S_{water}}{S_{\max}}\times 100=\frac{1-e^{-\frac{540}{560}}}{1+e^{-\frac{540}{560}}}\cdot \frac{540}{610}\times 100=39.6$$

Surface wind erosion rate, monthly corrosion rate of Q235 steel:(12)$$Z_{C_{water}}=\frac{S_{\min}}{S_{water}}\times 100=\frac{1.359}{2.265}\times 100=60$$
(13)$$Z_{G_{water}}=\frac{S_{\min}}{S_{water}}\times 100=\frac{0.081}{0.192}\times 100=42.2$$

Annual average use cost per unit area, acute oral LD50 toxicity classification in mice:(14)$$Z_{D_{water}}=\frac{S_{\max}-S_{water}}{S_{\max}-S_{\min}}\times 100=\frac{2000-100}{2000-100}\times 100=100$$
(15)$$Z_{F_{water}}=\frac{S_{\max}-S_{water}}{S_{\max}-S_{\min}}\times 100=\frac{6-1}{6-1}\times 100=100$$

pH value:(16)$$Z_{E_{water}}=\frac{100}{\sqrt{2\pi }}\cdot \exp^{-{\left(S_{water}-7\right)}^{2}}=\frac{100}{\sqrt{2\pi }}\cdot \exp^{-{\left(7-7\right)}^{2}}=100$$

The final calculation result is shown in Figure 8. According to the results in Figure 8, it can be directly seen that, compared with purified water, calcium chloride, Triton X-100, and polyacrylamide, the wetting performance of Triton X-100 is the best. The hygroscopic performance of the four chemical solutions is similar, and the performance of Triton X-100 is more prominent; the bond performance of polyacrylamide is the best; the comprehensive cost of purified water is the least; the pH value of the four selected chemicals is similar; the chemical toxicity of purified water is the least; polyacrylamide has the lowest chemical corrosion. Substituting the calculation results in Figure 8 into Equation (8), combined with the weight determination results of the evaluation parameters in Table 2, the comprehensive evaluation scores of the dust suppression performance of the four chemical dust suppression solutions are shown in Table 4.

According to the comprehensive dust suppression performance score results shown in Table 4, the comprehensive dust suppression performance score of Triton X-100 solution is in the poor-grade category. The comprehensive dust suppression performances of calcium chloride solution, water, and polyacrylamide solution scored high in the average-grade category. The comprehensive evaluation process is logically correct, and the results are consistent with the phenomena observed in the experiment, consistent with conventional understanding, and have strong credibility

## 6. Conclusions

A comprehensive chemical dust suppressant performance evaluation index system was constructed based on the dust suppression mechanisms, the cost-effectiveness, and environmental safety performance. Dust suppression performance comprises the wetting performance, hygroscopic performance, and bond performance. The economic indicator is the comprehensive use cost. Indicators of environmental safety are the pH value of the dust suppression solution, chemical toxicity, and chemical corrosion. The specific evaluation parameters are further determined, which are sedimentation and wetting time, evaporation stabilization time, surface wind erosion rate, annual average use cost per unit area, pH value of dust suppression solution, acute toxicity classification of chemicals, monthly corrosion rate of Q235 steel, and the data of each evaluation index were standardized by different methods according to the characteristics of the different indexes to enable comparison of the results.

According to the specific requirements of the evaluation index system, the corresponding standardized test method of evaluation parameters is proposed: the sedimentation wetting time adopts the standardization method combining the linear proportion of the reverse index and the range transformation method; the evaporation stabilization time adopts the linear proportion transformation method of the positive index, and uses the efficacy coefficient to adjust the range of standardized values; the surface wind erosion rate and the monthly corrosion rate of Q235 steel adopt the linear proportional transformation method of the reverse index; the pH value of the solution was evaluated by the moderate index transformation method; the annual average use cost per unit area and the grading of acute LD50 toxicity in mice were determined by the inverse index range transformation method. This evaluation system provides a reference method for the optimization and performance comparison analysis of chemical dust suppressants.

The introduction of the weight of the proposed evaluation system is conducive to the flexible adjustment of specific demands for dust suppressants without application scenarios; hence, the evaluation method has wide applicability and practicability.

The experimental test data obtained through the example show that: among the four chemicals selected to participate in the experiment, the comprehensive dust suppression performance score of Triton X-100 solution is in the poor-grade category. It also shows that the evaluation method is operable and reliable, providing a set of standardized evaluation methods and workflows for the optimization of dust suppressants.

## Figures and Tables

**Figure 1 ijerph-19-05617-f001:**
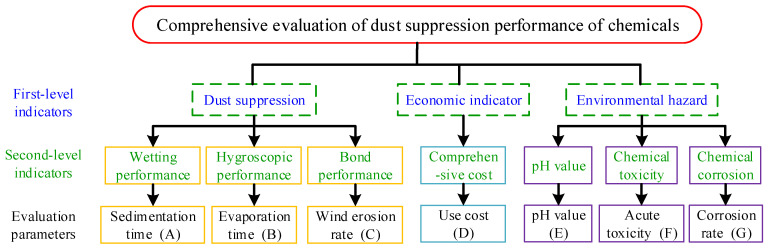
Schematic diagram of comprehensive evaluation index system.

**Figure 3 ijerph-19-05617-f003:**
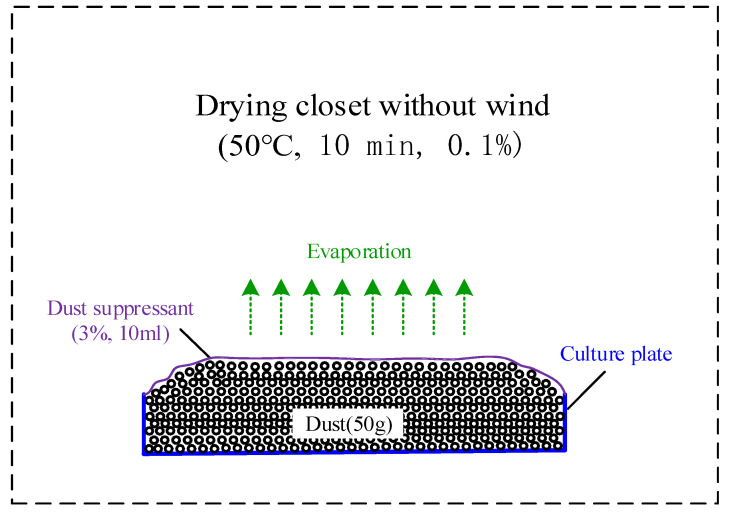
Schematic diagram of the stabilization time experiment.

**Figure 4 ijerph-19-05617-f004:**
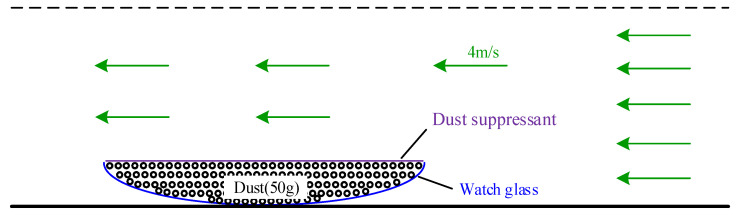
Schematic diagram of surface wind erosion rate experiment.

**Figure 5 ijerph-19-05617-f005:**
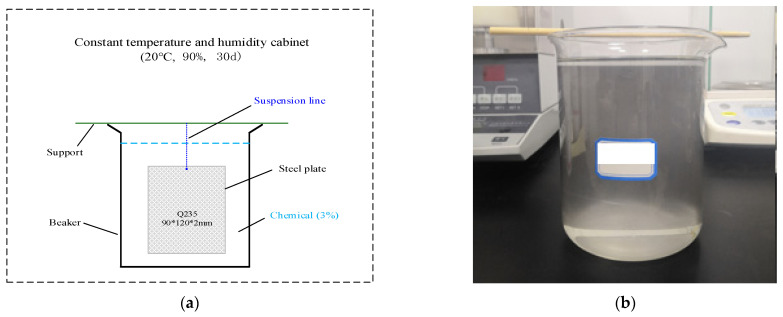
Monthly corrosion rate of Q235 steel experiment. (**a**) The experimental schematic diagram; (**b**) The experimental figure.

**Figure 6 ijerph-19-05617-f006:**
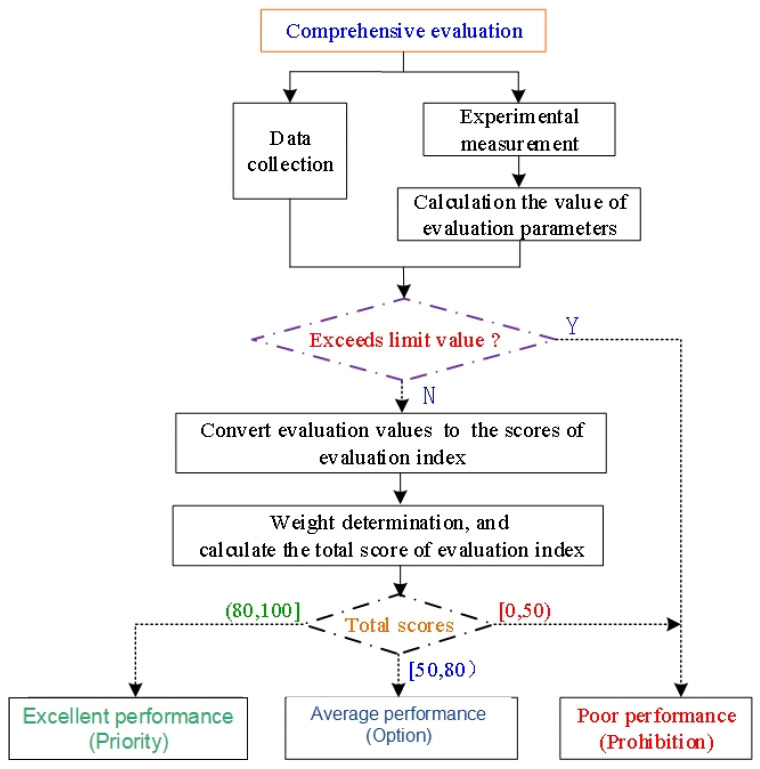
Flowchart of comprehensive evaluation of optimized dust suppressant.

**Figure 7 ijerph-19-05617-f007:**
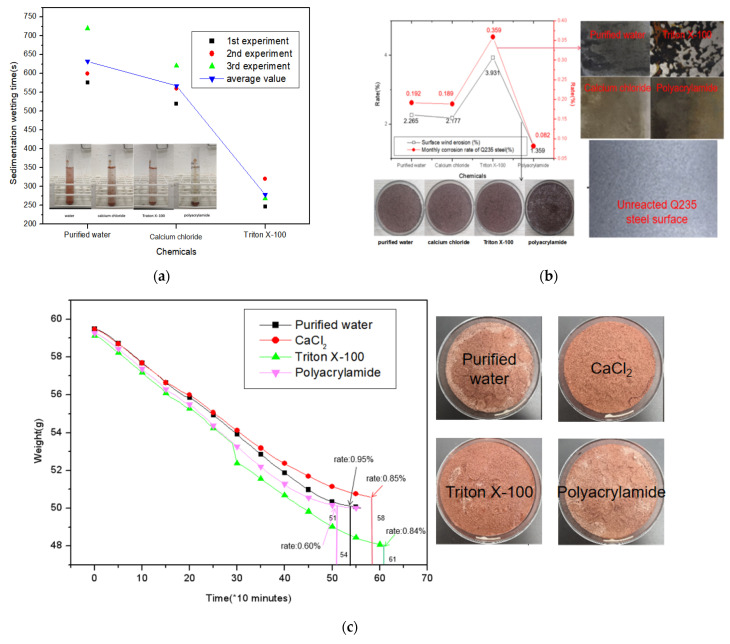
Process diagram of evaluation parameter measurement experiment. (**a**) Distribution map of experimental data for sedimentation wetting time measurement; (**b**,**c**) Evaporation stabilization time measurement experimental phenomenon and the trend graph of the data changing with time.

**Figure 8 ijerph-19-05617-f008:**
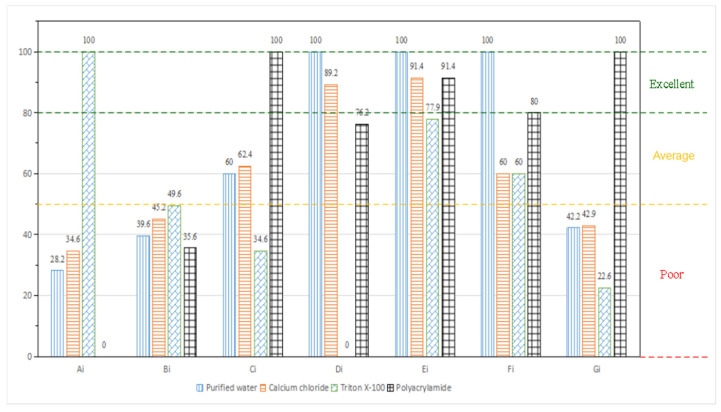
The statistical figure of standard values of several chemical materials for dust suppression.

**Table 1 ijerph-19-05617-t001:** Technical specification for chemical toxicity identification.

Toxicity Classification	Toxicity	LD50 (mg/kg)
Grade 6	Extremely toxic	<1
Grade 5	Highly toxic	1–50
Grade 4	Moderately toxic	51–500
Grade 3	Slightly toxic	501–5000
Grade 2	Practically non-toxic	5001–15,000
Grade 1	Non-toxic	>15,000

**Table 2 ijerph-19-05617-t002:** Table of valuation parameter weight coefficients.

	*W* _III1_	*W* _III2_	*W* _III3_	*W* _III4_	*W* _III5_	*W* _III6_	*W* _III7_
C	0.2	0.2	0.2	0.1	0.05	0.05	0.2

**Table 3 ijerph-19-05617-t003:** Evaluation parameter data of several chemical materials for dust suppression.

	Purified Water	Calcium Chloride	Triton X-100	Polyacrylamide
Sedimentation wetting time (s)	632.0	566.7	278.3	1257.6
Evaporation stabilization time (min)	540	580	610	510
Surface wind erosion rate (%)	2.265	2.177	3.931	1.359
Annual average use cost per unit area (yuan)	86	312	2184	585
pH value of dust suppression solution	7.0	6.5	7.3	6.7
Acute toxicity classification of chemicals (mg/kg)	Grade 1 -	Grade 3 1000 *	Grade 3 3500 *	Grade 2 12,950 *
Monthly corrosion rate of Q235 steel (%)	0.192	0.189	0.359	0.081

* indicate the reference values of the literature [17].

**Table 4 ijerph-19-05617-t004:** Comprehensive evaluation score of dust suppressant performance of several chemical materials.

Chemical	Purified Water	Calcium Chloride	Triton X-100	Polyacrylamide
Score	54.00	53.51	48.25	63.31

## Data Availability

Correspondence and requests for materials should be addressed to Rujia Wang.

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
