# Peer review of "Comprehensive Chemical Dust Suppressant Performance Evaluation and Optimization Method"

_ijerph, 2022, doi:10.3390/ijerph19095617_

Round 1
Reviewer 1 Report
The paper resulted well structured and I have just some concerns
Generally, the work is too descriptive and too little practical
In section 3 Explain better what you mean with a test solution configured according to the mass concentration of 3%
In section 5 explain better the case analysis, with more details, highlighting the successive steps.
How many times was the experiment repeated?
Also the legenda of figure 7 must be more explicative.
The comments to the figure 8 are poor
Why in the legenda of table 3 did you Write “Table 3. This is a table”?
The conclusion must be more critical
Reviewer 2 Report
This interesting manuscript needs improvement before it can be considered for publication
--Improve the abstract - expand and provide more details
--avoid lumping references. Discuss each individually or don't include
--more detailed figure captions are required. If there are multiple figures include (a) (b) etc.
--Discuss figures in detail in the manuscript
--are all variables defined?
--all equations need a number...if multiple use (a), (b) etc
--improve english grammar and paper structure (i.e word choice, sentence structure , readability, etc)
Reviewer 3 Report
The article 'Comprehensive Chemical Dust Suppressant Performance Evaluation and Optimization Method' provides valuable information on dust suppression technologies and their advantage and limitations. However, the authors must address the following comments-
The background and significance of this study needs to be thoroughly established by providing more relevant background information so that the readers can identify the problem areas. Providing data related to health and environmental issues would be an ideal way to address this particular deficiency.
All figure and table captions must be detailed. Each and every terms and symbols need to be explained.
Figure 7b: There is a typo. Traton X-100 should be Triton. Also, the authors must mention the instruments used for magnification and its value.
This article is missing the discussion part significantly. The authors must describe the results in details as well as provide scientific justifications for the findings.
Round 2
Reviewer 2 Report
This manuscript has been improved, however it still requires some improvement.
--There are still several lumped references. Please discuss each reference individually.
--Please provide detailed captions for all figures and tables.
--Some discussion has been added to this revision. Please provide additional discussion of the figures.
--Improve English grammar and paper structure as previously described. There has been some improvement to this version, however additional improvement is still required.
--
